# Online Learning for Right-Sizing Serverless Function Invocations

Prasoon Sinha*, Kostis Kaffes†, Neeraja J. Yadwadkar*‡,

*The University of Texas at Austin. *prasoon.sinha@utexas.edu, neeraja@austin.utexas.edu*

†Google SRG. *kkaffes@gmail.com*

‡VMware Research.

*Abstract*—Serverless computing relieves developers from the burden of allocating and managing resources for their cloud applications, providing ease-of-use to the users and the opportunity to optimize resource utilization to the providers. However, the lack of visibility into user functions limits providers' ability to right-size the functions. Thus, providers resort to simplifying assumptions, ignoring input variability, and coupling different resource types (CPU, memory, network), resulting in widely varying function performance and resource efficiency. To provide users with predictable performance and costs for their function executions, we need to understand how these factors contribute to function performance and resource usage.

In this paper, we first conduct a deep study of commonly deployed serverless functions on an open-source serverless computing framework. Our analysis provides key insights to guide the design of a resource allocation framework for serverless systems, including the need to provision resources per invocation, account for function semantics, and decouple resources. We then present *Lachesis*, a resource allocation framework that builds on the insights we found and leverages online learning to right-size a function invocation. Our experiments show that Lachesis can increase speedup by 2.6x while decreasing idle cores by 82% compared to static allocation decisions made by users.

## I. INTRODUCTION

A key benefit of serverless computing for users is that they get to focus on their application logic and leave the details of resource provisioning and management to the cloud providers. However, this results in an opaque interface between users and providers that adversely impacts both. For users with performance-critical applications, such as timely detection of videos with indecent content uploaded to YouTube, or cost-minded applications, such as personal photo organization, unknown resource management policies that they cannot control are a problem [12], [19]. Meanwhile, providers lack visibility into user-functions limiting their ability to make cost-performance trade-offs on behalf of the users.

Existing serverless systems either completely hide the resource allocation policies they use [16], or provide a single knob, the memory size of the container, that the user can set [5], [10]. This parameter is intended to give users control over resource management and providers visibility into the resource requirements of user functions. However, even with this additional input, serverless systems are incapable of providing performance- and cost-aware function execution to users. We argue that, to fix this issue, we need to first understand which factors impact function performance and how. We then need to study how the current resource allocation frameworks

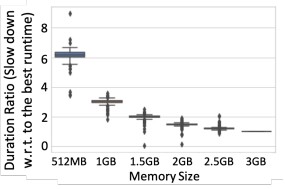

(a) Slowdown w.r.t. the best runtime across mem sizes for 100 invocations of a video transcoding function.

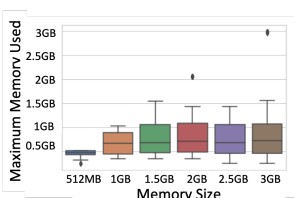

(b) Maximum mem utilized vs. allocated across 100 runs of the video transcoding function (from Fig. 1a).

Fig. 1: Characterizing functions with respect to the resources allocated, utilized, and performance observed.

take these factors into account. Finally, we need to see the combined impact of existing policies and these factors on function performance and resource efficiency. We review the following policies and assumptions made by existing resource allocation frameworks for serverless systems and motivate the need for our characterization work.

**1. Static and input-agnostic allocation by providers:** Providers statically allocate resources to functions using the user-specified memory size. However, this approach ignores the fact that different inputs submitted to the same function might have different resource needs (as demonstrated by the spread in duration in Figure 1a). This precludes optimizations such as using smaller containers for smaller inputs, which might bring significant cost benefits. For instance, if the static function-level allocator sized a container with 3GB memory, and most of the invocations used only 1GB, the costs incurred are 2x higher for allocated but unused resources. *Hence, we need to understand the impact of inputs and function semantics on function performance and resource utilization.*

**2. Coupled allocation of different resource types by providers:** The specified memory size for a function dictates the number of CPU cores, thus, tightly coupling the two types of resources together. There are two main limitations with this approach: (a) Although users now need to only tune the memory knob, setting this knob correctly might be difficult for workloads that are not memory-intensive but limited by other resources. For instance, video transcoding or compression are CPU intensive workloads. Users might have to profile their functions carefully, adding significant cost. (b) The tight coupling of resources might lead to suboptimal resource allocation decisions for certain kinds of workloads. For example,

| Function | Input Type | # Runs | # Sizes | Size Range |
|---|---|---|---|---|
| *matmult* | square matrix | 540 | 9 | 500 - 80000 |
| *linpack* | square matrix | 660 | 11 | 500 - 8000 |
| *image-process* | image | 840 | 14 | 12K - 4.6M |
| *video-process* | video | 645 | 5 | 2.2M - 6.1M |
| *encrypt* | string | 420 | 7 | 500 - 50000 |
| *mobilenet* | image | 840 | 14 | 12K - 4.6M |
| *sentiment* | batch of strings | 716 | 12 | 50 - 3000 |
| *speech2text* | audio | 471 | 8 | 48K - 12M |
| *qr* | url | 660 | 11 | 25 - 480 |
| *lr-train* | training set | 160 | 4 | 10M - 100M |
| *compress* | file | 434 | 7 | 64M - 2G |
| *resnet-50 (inf)* | image | 574 | 9 | 184K - 4.6M |

TABLE I: Summary of 12 serverless functions studied.

CPU intensive workloads might end up being allocated large amounts of memory that are not used (Figure 1b).

**3. Over-provisioning by users:** The importance of the memory size parameter [6] and its opaque coupling to other resources forces users to profile their function to find the right setting. But, as the performance and resource usage of a function can depend significantly on inputs, users must profile on diverse inputs to ensure adequate resource availability in all cases. The cost of doing so is prohibitive. Prior work on reducing this profiling cost either assumes knowledge of the workload which is unavailable [4], [18], or ignores the input itself, which can have a large impact on many functions [20]. Thus, users often overprovision and choose the largest memory size available (10GB for AWS Lambda, for instance), raising their costs significantly and leading to underutilized resources for the provider [2]. *Hence, we need to understand how inputs affect a function's resource demands.*

In this paper we extensively study the impact of function inputs and resource coupling on several serverless functions covering a wide range of application types. Building on the insights we found, we introduce *Lachesis*, an online learning based resource allocation framework that (1) allocates resources to each function invocation *based on characteristics of the input and function semantics*, and (2) *decouples* different resource types. Lachesis employs an online learning agent that uses cost-sensitive multi-class classification to predict the minimum number of cores required to satisfy a given invocation's service level objective (SLO). It removes the need for users to specify memory limits, and in doing so, Lachesis achieves betters resource utilization while simplifying the serverless user interface.

## II. EXISTING RESOURCE ALLOCATION MECHANISMS

Several cloud providers, such as AWS Lambda [5] and Google Cloud Functions [10], and open-source communities [17] expose a common interface to their serverless platforms: users specify a memory limit for their functions at creation time. The platforms then allocate a proportional amount of CPU based on the memory limit. Thus, all invocations of a function have the same container size, regardless of their actual resource needs. Apache OpenWhisk's [17] CPU allocation is a soft limit, as invocations can burst if there are available CPUs in the server. Microsoft Azure [16] claims to automatically scale the resources allocated to functions, but its resource

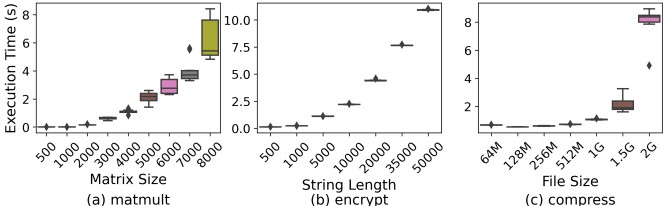

Fig. 2: Execution time as a function of data size for three serverless functions. The CPU and memory limit is fixed across sizes.

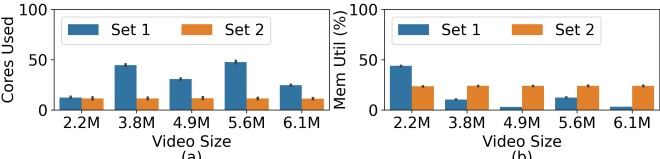

Fig. 3: *video-process*'s (a) CPU and (b) memory utilization as a function of video size. The CPU limit is fixed at 80 cores.

allocation policies are unknown, making it difficult to reason about function performance. Cypress [7] creates containers with high CPU count and memory capacity per function to consolidate multiple concurrent function invocations within one container to avoid wasting resources. Bilal et al., [8] propose to decouple memory and CPU to create a trade-off between performance and cost. ReSC [11] divides functions into resource components (i.e., compute or memory) and allocates resources per component.

## III. WHAT AFFECTS FUNCTION PERFORMANCE?

We study the impact of input properties (i.e., size, type), resource availability, and coupling of resource types, on the performance and resource utilization of serverless functions.

**Experimental Setup:** Our study observes functions on OpenWhisk [17]. We make two changes to OpenWhisk. (1) We force all CPU limits to be hard limits. (2) We decouple CPU and memory to explore different configurations than the fixed pairings provided by OpenWhisk. We deploy OpenWhisk on two bare-metal nodes in TACC's Chameleon cluster [14]. Each node contains 2 AMD EPYC 7763 CPUs, operating at 2.45 GHz [1], and 251GB of memory. For performance predictability, we disable hyperthreading, as done in [13], resulting in 128 online cores per machine. We install Ubuntu LTS 18.04. One machine hosts the OpenWhisk Controller and CouchDB while the other hosts the Invoker to run functions.

**Workloads:** We study 12 functions (see Table I) from literature and benchmark suites [7], [9], [15] covering scientific applications, data processing, and machine learning (ML) inference serving and training. We collect the execution time and memory/CPU utilization for several combinations of functions, input sizes, and CPU limits. We run each combination 8-10 times, for a total of ~8K runs.

### A. Impact of Function Inputs

We study two questions: (1) What impact does input size have on function performance? (2) Do input properties, other than size, affect function performance and resource utilization?

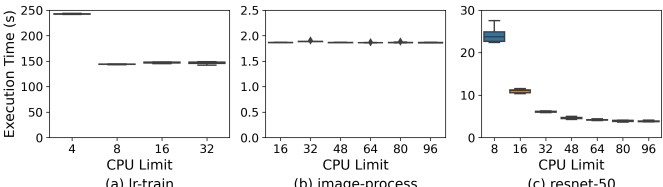

Fig. 4: Execution time as a function of CPU limit for three of our serverless functions. The input size is fixed at max value.

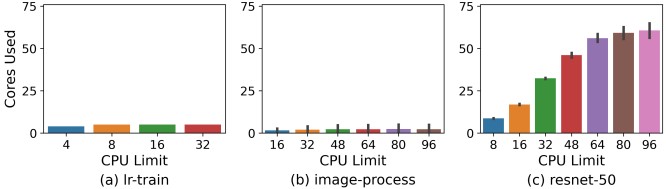

Fig. 5: CPU utilization compared to allocation for three of our serverless functions. The input size is fixed at max value.

**Observations:** Figure 2 presents input size vs. execution time for three functions (we omit the others due to space constraints). We note that regardless of input type (matrix, files, text) or function semantics (i.e., single- vs. multi-threaded), function *performance is correlated with input size and depends on whether the function is single or multi-threaded*.

Figure 3a compares the number of cores used by *video-process* on two input sets of different videos. We see that two inputs of the same size may vastly differ in the number of cores used. We also notice that while set-1 has an unpredictable relationship between input size and cores used, set-2 exhibits constant utilization regardless of video size.

To understand these differences, we compare video properties beyond just size: frame rate per second, video length, bit rate, and video resolution. We find that the resolution is the key property affecting resource utilization. While the resolution is constant in set-2 (1280 × 720), it widely varies between the different video sizes in set-1. Inputs with higher resolutions (1280 × 720) have lower CPU and higher memory utilization, whereas the inverse is seen for lower resolution inputs.

**Insights:** Function semantics and input properties (not just limited to size) affect performance and resource utilization. Existing resource allocators that ignore input properties beyond size are thus suboptimal. Instead, functions can benefit from allocators that account for inputs and function semantics.

### B. Impact of Added Resources

We now evaluate the effect of adding resources to a function. **Observations:** Figures 4a and 4c show that *lr-train* and *resnet-50* can benefit from more cores (execution time decreases). *matmult*, *compress*, and *linpack* also exhibit these trends. However, *lr-train* shows that the gains of increasing CPU saturate: execution time does not improve beyond 8 cores. In fact, Figure 5a shows that utilization never surpasses 5 cores. *lr-train* uses scikit-learn's LogisticRegressionCV() with *n_jobs*=-1 to implement training. This setting specifies to use as many cores as possible. Since *lr-train* does not specify the number of cross-validation folds, 5 folds (the default) are

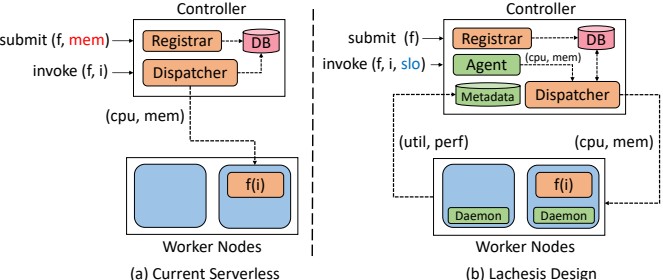

Fig. 6: (a) Current serverless platforms vs (b) Lachesis.

in the loop for training, and thus at most 5 cores are fully utilized.

Meanwhile, Figure 4b shows that *image-process* does not benefit from more cores, even though its performance is input-dependent as explained in § III-A. Figure 5b shows that regardless of CPU allocation, utilization is always hovering around 1 core. In fact, several of our functions are single-threaded: *mobilenet*, *sentiment*, *encrypt*, and *speech2text*.

**Insights:** Serverless platforms see a mix of single- and multi-threaded functions with potentially bounded parallelism. Adding resources may not always help. Hence, resource allocators should tailor their policies to suit the type of function.

### C. Impact of Coupled Resource Types

Existing allocation policies scale CPU in proportion to the user-specified memory size [5]. However, this assumes functions are both CPU- and memory-intensive. Here, we evaluate the accuracy of this assumption.

**Observations:** Figure 3 shows that *video-process* uses up to 50 cores, but its memory utilization is at most 41% (0.8GB). Thus, *video-process* (also *matmult*, *linpack*, and *lr-train*) is compute-intensive. Conversely, we found *sentiment* to be memory-bound (100% memory utilization while it uses at most 1 core). Thus, *different functions may utilize resource types in different proportions*. Cloud providers can experience severe underutilization due to resource coupling. For example, providing enough memory to *sentiment* would lead to 50% underutilization of allocated vCPUs. Meanwhile, to allocate 50 vCPUs to *video-process* would require a 88GB memory allocation, resulting in ∼99% memory underutilization.

**Insights:** It is imperative that allocators decouple resources to improve utilization while meeting resource demands.

### IV. LACHESIS DESIGN AND IMPLEMENTATION

We now present Lachesis, a system that makes fine-grained and decoupled resource allocations per invocation using an online learning agent. Figure 6a shows a simplified architecture of existing serverless systems. Figure 6b shows the changes we make to the existing workflow of serverless frameworks: Lachesis simplifies the user interface by removing the need for users to specify a static memory limit during function submission. Instead, users can simply provide an SLO *per invocation*. Given a function, input, and SLO, Lachesis aims to right-size invocations by dynamically allocating the minimum amount of resources to meet the SLO.

---

**Algorithm 1** Lachesis' logic using online learning

**Input** $fxn$, $in$, $slo$
1: Determine $cpu\_lim$: default or ModelPredict($in$, $slo$)
2: Launch $fxn$ with given $in$ and determined $cpu\_lim$
3: Observe $fxn$'s $max\_cpu$ and $exec\_time$ during runtime
4: Use $max\_cpu$ and $exec\_time$ to ComputeCosts()
5: Update online model: ModelUpdate($fxn$, $in$, $slo$, $costs$)

---

Algorithm 1 summarizes Lachesis' logic. We focus on CPU allocation in this paper and leave memory allocation as future work. For an invocation, the online learner predicts the minimum number of cores to allocate. Lachesis defaults this value if the learner has not seen enough invocations for the given function. It then launches the invocation with the determined CPU limit and collects utilization and duration metrics for feedback. Finally, it uses the observed data to compute costs and updates the online learner after every invocation. Next, we describe the formulation of our online learning agent.

**Prediction Target**: As our goal is to meet user-specified SLOs with efficient use of resources, a natural prediction target is the minimum number of cores a function invocation needs for a target execution time.

**Model Inputs**: Our model's inputs are the serverless function, user inputs, and an SLO. We built a function and input featurizer that automatically extracts features from functions and inputs. We extract function features that can potentially impact its performance, such as the number of function calls, libraries used, and loop sizes. Unlike functions, we extract different features for different input types. For example, for images we extract the image's file size and resolution, whereas for a matrix we extract its size and density. We combine all this data to construct a vector for model updating and prediction.

**Feedback**: On each worker machine, we deploy a daemon that captures the maximum CPU utilization over the invocation's runtime. This data is used by our cost function to update our model's weights online.

**Learning Algorithm**: We approach predicting core count as a supervised learning problem, which can be solved with regression or classification. We opt to not use regressors because of the difficulty in formulating a cost function to differentiate between underpredictions and overpredictions upon an SLO violation. Instead, we use *cost-sensitive multi-class classification* to make predictions. Each class (core count) has its own linear regressor that predicts the class's cost for an invocation. We select the class with the lowest cost as the allocation. Now, we can differentiate costs for different classes without worrying about the relationship between them.

**Cost Function**: Our cost function is rather intuitive. First, we determine the class to assign the lowest cost of one to. There are three cases. (1) If an invocation's SLO is met, the $max\_cpu$ (i.e., the maximum number of cores used by the invocation) class is given the lowest cost. Hence, if allocated resources are not efficiently utilized, our agent can learn to make smaller allocations for similar future invocations. (2) If

an invocation's SLO is met and all assigned cores are used, we may assign a class lower than $max\_cpu$ the lowest cost. This class is determined based on the slack between the invocation's execution time and SLO. In doing so, we inform our online learner that fewer cores may also satisfy this invocation's SLO. (3) Upon an SLO violation, we assign a class greater than $max\_cpu$ (at most $+10$) the lowest cost in an attempt to meet the SLO in the next invocation. Similar to case (2), the slack determines this class. After determining the lowest cost class, the costs of the remaining classes grow linearly, with underpredictions being penalized further by a hyperparameter.

**Implementation**: We implement Algorithm 1 as a shim layer that can run on top of any serverless platform. This layer runs on the same node as our dispatcher. We use Apache Open-Whisk [17] as our base serverless platform and implement our online learning agent using Vowpal Wabbit [3], a library with an efficient online implementation of the cost-sensitive multi-class classification algorithm. On each Invoker, we launch a metric aggregation daemon that collects utilization and runtime metrics per invocation and persists the data in a Metadata store for the shim layer to use when updating its models.

**Why Online Learning**: The fundamental limitation of existing public serverless platforms [5], [10] is their inability to right-size containers dynamically based on inputs. Meanwhile, for Cypress to achieve high utilization, arrival patterns need to be frequent enough to pack invocations in one container within the window of an SLO [7]. Hence, Cypress is susceptible to severe resource underutilization with sparse resource arrival patterns. Finally, as shown in § III, it is infeasible to use heuristics to predict optimal resource allocation because of variation in function behaviors depending on function semantics and input types/properties. This prompts our use of online learning, enabling Lachesis to dynamically right-size containers and adapt to changes in function and input distribution over time.

## V. Evaluation

We aim to show Lachesis' efficacy in allocating resources per invocation. Specifically, we evaluate the impact of per-invocation allocations on the number of SLO violations, resource utilization, and user cost.

### A. Methodology

**Baselines**: We compare Lachesis to three baselines, on Open-Whisk (ow), users might choose when providing resource needs to existing serverless platforms. Users may ask for the maximum, median, or minimum amount of resources for all their invocations. These correspond to our ow-large (64 cores), ow-medium (32 cores), and ow-small (1 core) baselines.

**Workloads**: We evaluate Lachesis with three serverless functions from Table I: *image-process*, *matmult*, and *resnet-50*. While *image-process* is single-threaded, both *matmult* and *resnet-50* are multi-threaded, showing the robustness of our system to both types of functions. For each function, we run over 100 invocations with over 60 different inputs for *image-process* and 20 for both *resnet-50* and *matmult*. The trace of invocations is the same on Lachesis and our three baselines.

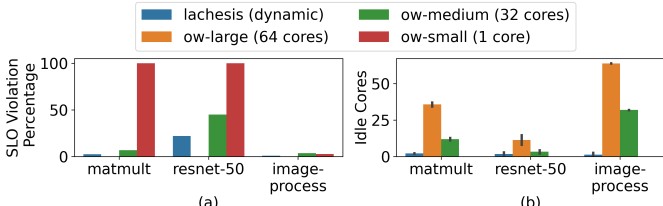

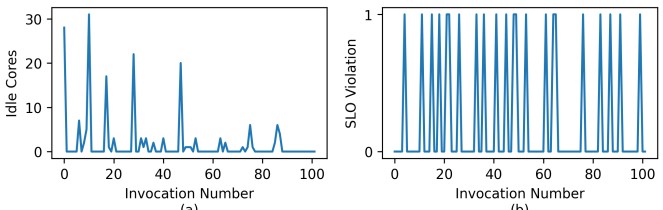

Fig. 7: Difference in (a) SLO violation percentage and (b) idle cores between Lachesis and our baselines for three functions.

Fig. 8: A timeline view of Lachesis' number of unused cores (blue) and SLO violations (green) over 100 invocations of *resnet-50*.

**Evaluation Metrics**: Lachesis aims to meet an invocation's SLO using the minimum number of cores it can. Hence, we are interested in two metrics: a function's SLO violation ratio and CPU utilization. (1) Each function input has its own SLO (max execution time). We determined this value by profiling each input with different allocations and extracting the best execution time we could achieve. We then increased this value by 10% and considered this the input's SLO. A function's SLO violation ratio is the number of SLO violations to the number of invocations. (2) We report CPU utilization as the number of idle allocated cores. This is because a 50% underutilization using $\frac{1}{2}$ cores is not as severe as using only $\frac{16}{32}$.

## B. Results

We compare the SLO violations (Figure 7a) and CPU utilization (Figure 7b) between Lachesis and the three baselines. Our baselines display an inherent tradeoff between meeting invocation SLOs and achieving optimal CPU utilization. While ow-large meets all SLOs, resource utilization is poor, as most inputs do not require an allocation of 64 cores. Meanwhile, ow-small is unable to meet any of the SLOs (100% violation) for our multithreaded-functions (*matmult*, *resnet-50*), but achieves perfect CPU utilization because every function uses at least 1 core. The ow-medium baseline allocates 32 cores to all invocations. While 32 cores are enough for many invocations, there are still plenty of inputs that require more than 32 cores to meet the SLO. Lachesis dynamically learns the minimum core count required to meet the SLO, thereby reducing the number of idle allocated cores while decreasing the number of SLO violations compared to ow-medium. This translates into a significant impact on user cost, as for *resnet-50* alone Lachesis reduces cost by 63% for 100 invocations.

Figure 8 shows Lachesis' number of idle cores and SLO violations over the course of 100 invocations of *resnet-50* with various inputs. It takes 28 invocations for Lachesis to stabilize and learn the minimum number of cores required for different inputs. For the remaining invocations, the number of idle cores is less than 8, except for one spike at invocation 47. Interestingly, throughout the course of the 100 invocations, there continues to be periodic SLO violations. We noticed that these violations are for the same input, which had an unrealistic SLO. For each invocation of this input, Lachesis would allocate more cores in an attempt to meet the SLO, however even with the max 64 cores, the SLO was never met.

## VI. CONCLUSION

For ease-of-use and resource efficiency of serverless platforms, our analysis motivates that resource allocation should be fine-grained per invocation and per resource type, to account for various input properties. We present Lachesis that uses an online learner to predict the number of cores required to meet an invocation's SLO and show its efficacy in improving performance, resource utilization, and user cost.

**Future Work:** Lachesis paves the path for the following next steps: (1) Currently, Lachesis creates one online agent per function due to the variable number of features extracted from different input types (e.g., video, audio). We plan to standardize features to enable a single agent to allocate resources for all functions. (2) Lachesis decouples resource types, but currently focuses on only making CPU allocations. We will augment Lachesis by allocating memory per invocation as well. (3) While per-invocation allocations help as we demonstrated in this paper, customized allocations per invocation also increase the number of containers used per function, thereby increasing the number of cold-starts. Cold-starts often worsen function performance. We will design a scheduler that closely interacts with our resource allocator to strike the right trade-off between improved utilization due to fine-grained allocations and resulting cold-starts.

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
