# OpenReview forum: "Online Learning for Right-Sizing Serverless Functions"
_iscaconf.org/ISCA/2023/Workshop/ASSYST — ASSYST Oral_

### Official Review · Reviewer_TgJJ · 2023-05-05
**Good Paper. Well-researched and well-written. Proposes an approach to right-sizing serverless function invocations using online learning.**

**Rating:** 7
**Confidence:** 3

**Review:**

The paper addresses a significant issue in serverless computing, where the lack of visibility into user functions limits providers' ability to right-size the functions. This leads to unpredictable performance and cost for function executions, which can be a major problem for users.

The paper includes a good analysis of commonly deployed serverless functions on an open-source serverless computing framework and provides valuable insights into factors that impact function performance and resource usage.

The authors propose a solution to this problem by introducing a resource allocation framework called Lachesis that leverages online learning to right-size a function invocation. Experimental results show that Lachesis can increase speedup by 2.6x while decreasing idle cores by 82% compared to static allocation decisions made by users.

The paper is well-structured and easy to understand. The authors have provided clear explanations of their methodology and experimental results. However, the authors could provide more details on the limitations of their research, the potential impact on cost, and future research directions.

**Review (Strengths/Weaknesses):**

Strengths:
1. The paper addresses an important issue in serverless computing: the lack of visibility into user functions limits the providers' ability to right-size the functions. They propose a solution that can provide users with predictable performance and costs for their function executions.
2. The authors have provided a good analysis of commonly deployed serverless functions on an open-source serverless computing framework, providing valuable insights into the factors that impact function performance and resource usage.
3. They have presented a resource allocation framework called Lachesis that leverages online learning to right-size a function invocation. They provide experimental results that show Lachesis can increase speedup by 2.6x while decreasing idle cores by 82% compared to static allocation decisions made by users.
4. The paper is well-structured and provides a clear overview of the research problem, methodology, and experimental results.

Weaknesses:
1. The authors could provide more details on the limitations of their research, such as potential biases and assumptions made in their analysis or experimental setup.
2. They could discuss the potential impact of their framework on the cost of running serverless applications.
3. The paper could benefit from more extensive experiments to evaluate the effectiveness of their framework in a broader range of scenarios.
4. The authors briefly mention future work at the end of the paper but could provide more details on potential directions for future research.

**Reviewer Expertise:**

Little or no familiarity.

---

### Official Review · Reviewer_iP5R · 2023-05-06
**The paper tackles a real problem which is resource right-sizing in serverless systems with an approach that is aligned with the serverless design paradigm. There is however, room for improvement in setting up the right motivation and details of evaluation.**

**Rating:** 4
**Confidence:** 4

**Review:**

Summary:
The paper suggests an online learning algorithm to find the right amount of resources for serverless containers to reduce idle resources while meeting the user SLOs.

Pros:
- Serverless functions are aimed in reducing developer effort in managing the resources and delegating that to the platform. The motivation of the paper is aligned with this design goal. The paper suggests delegating the right-sizing task to the platform so the developer does not need to explore this design space.
- Serverless platforms are beneficial in the unpredictable environment, i.e., when the qps is not constant or pre-determined. A dynamic approach for right-sizing is aligned with this benefit of serverless functions
- The resource right-sizing can bring dollar value to the vendors by increasing resource efficiency.
- A white-box approach as presented in the paper can be insightful for the algorithm and can potentially results in better resource efficiency, compared to a block-box algorithm.


Cons:
- On the other hand, a white-box approach might not be well received by customers. It could also allow developer to manipulate the learner algorithm.
- There are currently some state-of-the-art serverless offerings from major vendors, e.g., Azure functions, that do not require user to specify the memory/cpu for the function. This is against the assumption/motivation of the paper.
- The on-line right sizing will impact the debug-ability of the serverless function. That is, if the SLO is not met, how the developer is supposed to debug their service to tackle the issue?
- A complicated algorithm such as on-line learner would bring computation and memory overhead to the serverless platform, hence negating the purpose. This overhead needs to be studied.

**Review (Strengths/Weaknesses):**

Strength:
- A dynamic approach seems to be suitable for the serverless environment, due to the unpredictable nature
- The paper is written clearly and well-structured. It's also very smooth to read.
- A supervised on-line learning algorithm seems to be the right choice and shown to be converging very quickly.

Weakness:
- The overhead of learner is not discussed in the paper. How much computation does the learner add and does the cpu saving compensate for that?
- Although the right-sizing problem is real and beneficial to the serverless vendor, but the motivation (section I) is not convincing. some vendors do not require users to specify memory
- they claim they decouple resources, however, they do not expand on it. The only focus of the results is on the cpu count and not much discussion on the memory sizing
- The trace qps and bursty-ness are important in evaluating the serverless functions, which is not considered in this paper
- Serverless containers are often re-used in the frequent invocations (called warm-start) to avoid the O(seconds) latency of starting a new container. The right-sizing approach, however, is only applicable when a new container is spawned. This can reduce the opportunities that the learner can impact the efficiency. The warm-start needs to be considered in this study
- The paper does not discuss how the function features can be extracted from the code. have they done this manually? do the authors suggest any specific approach to extract features from the code?
- Comparison with other dynamic approaches such as some black-box right-sizing algorithms is required.

**Reviewer Expertise:**

Expert: I have written one or more papers on this topic and/or I currently work in this area.

---

### Official Review · Reviewer_SkN5 · 2023-05-08
**The paper proposes a resource allocation framework using online learning for serverless functions, to help increase speedup while reducing the idle cores.**

**Rating:** 5
**Confidence:** 3

**Review:**

The paper first presents various cases to show how the static resource allocation scheme allocates cores to user functions in an open source serverless computing framework. This naive approach lacks the knowledge on the user functions such as input variability, interaction between various resource types, thereby, leading to a potential non-ideal allocation. The proposed framework, Lachesis, uses these insights to find the right number of cores to improve the utilization while minimizing the SLO violations. The evaluation is performed on three functions, image-process, matmul and resnet-50. The increase in speedup is 2.6x with an 82% reduction in idle cores.


**Review (Strengths/Weaknesses):**

The paper tackles an essential problem of optimal resource allocation, especially in serverless computing frameworks with various workloads. The initial set of experiments to unearth the issues with current schemes is clean. It can be more rigorous by including state-of-art approaches.

Is the frequency of Lachesis controller determined by the type of workload and its behaviour? Are there any overheads or limitations by running this on the same node as the dispatcher. What is the frequency of invoking the online learning component and what happens if a resource partition is found to be not working as efficient as expected?

Assuming the results in figure 8 are aggregated across the three functions, is there a difference in the results for single-threadedd vs. multi-threaded workloads. More rigorous evaluation with additional workloads would have been useful to understand the strengths of this framework.

**Reviewer Expertise:**

Knowledgeable: I used to work in this area and/or I try to keep up with the literature but might not know the latest developments.

---

### Official Review · Reviewer_5Hti · 2023-05-11
**Robust analysis, good solution, clear presentation, with some limitations**

**Rating:** 6
**Confidence:** 3

**Review:**

The paper addresses the challenge of dynamic resource allocation in serverless computing, which enables developers to focus on programming without concern for the underlying infrastructure. However, some serverless platforms do not permit users to specify resource requirements for their tasks, and most allow only the memory allocation to be specified. The arbitrary allocation of other resources, such as CPU, GPU, and network bandwidth, based solely on memory allocation can result in resource underutilization and increased costs. The study provides an in-depth analysis of commonly deployed serverless functions and identifies three critical issues with current resource allocation frameworks. The authors propose a novel machine-learning-based resource allocation approach that utilizes supervised online learning to allocate resources optimally to meet service level agreements (SLAs) at the lowest cost. The manuscript is well-written, and the proposed approach exhibits novelty and potential practical value.

**Review (Strengths/Weaknesses):**

Strengths points:
  * The paper is clearly written and well-presented, even for readers with limited background in the topic.
  * The analysis of various types and sizes of workload is robust.
  * The proposed framework shows excellent speedup improvements of up to $2.6\times$ and a decrease of $82$\% in the number of idle cores compared to the existing static allocation approach.
   * The proposed resource allocation framework demonstrates significant speedup improvements of up to$2.6\times$ and a decrease of $82$\% in idle cores compared to the existing static allocation approach.
  * The study includes a diverse range of real-world workloads, including machine learning inference and training applications.

Weaknesses:
  * Although the paper is well-presented and the proposed framework shows considerable cost reduction, the concept of dynamic resource allocation is not new and has been studied for many years, including the use of machine learning approaches to build similar frameworks.
  * While the paper presents a benchmark of 12 different functions that cover various applications, the final evaluation of the framework is limited to only three workloads that are similar in nature.
  * Although GPUs are increasingly popular for machine learning tasks, the paper does not provide an analysis of resource allocation for serverless systems that support GPUs or other custom resources such as TPUs on Google Cloud Functions. Additionally, there is no analysis of applications that require network resources.
  * The paper uses some abbreviations without providing clear definitions, which may present a challenge for readers who are not well-versed in the terminology of the topic.

**Reviewer Expertise:**

Knowledgeable: I used to work in this area and/or I try to keep up with the literature but might not know the latest developments.